# A Sensitive Micro Conductometric Ethanol Sensor Based on an Alcohol Dehydrogenase-Gold Nanoparticle Chitosan Composite

**DOI:** 10.3390/nano13162316

**Published:** 2023-08-12

**Authors:** Anis Madaci, Patcharapan Suwannin, Guy Raffin, Marie Hangouet, Marie Martin, Hana Ferkous, Abderrazak Bouzid, Joan Bausells, Abdelhamid Elaissari, Abdelhamid Errachid, Nicole Jaffrezic-Renault

**Affiliations:** 1Institute of Analytical Sciences, University of Lyon, 69100 Villeurbanne, France; anis19.work@gmail.com (A.M.); patcharapan.s@hotmail.com (P.S.); guy.raffin@isa-lyon.fr (G.R.); marie.hangouet@isa-lyon.fr (M.H.); marie.martin@isa-lyon.fr (M.M.); abdelhamid.elaissari@univ-lyon1.fr (A.E.); abdelhamid.errachid-el-salhi@univ-lyon1.fr (A.E.); 2Laboratory of Materials and Electronics Systems, University El-Bachir El-Ibrahimi Bordj Bou Arreridj, Bordj Bou Arreridj 34000, Algeria; a_bouzid34@hotmail.com; 3Center for Research and Innovation, Faculty of Medical Technology, Mahidol University, Nakhon Pathom 73170, Thailand; 4Laboratory of Mechanical Engineering and Materials, Faculty of Technology, University of Skikda, Skikda 21000, Algeria; hanaferkous@gmail.com; 5El Consejo Superior de Investigaciones Científicas (CSIC), Centro Nacional de Microelectrónica (CNM), Institut de Microelectrònica de Barcelona (IMB), Campus UAB, 08193 Barcelona, Spain; joan.bausells@imb-cnm.csic.es

**Keywords:** ethanol, mouthwash, alcohol dehydrogenase, chitosan, conductometry

## Abstract

In this paper, a microconductometric sensor has been designed, based on a chitosan composite including alcohol dehydrogenase—and its cofactor—and gold nanoparticles, and was calibrated by differential measurements in the headspace of aqueous solutions of ethanol. The role of gold nanoparticles (GNPs) was crucial in improving the analytical performance of the ethanol sensor in terms of response time, sensitivity, selectivity, and reproducibility. The response time was reduced to 10 s, compared to 21 s without GNPs. The sensitivity was 416 µS/cm (*v*/*v*%)^−1^ which is 11.3 times higher than without GNPs. The selectivity factor versus methanol was 8.3, three times higher than without GNPs. The relative standard deviation (RSD) obtained with the same sensor was 2%, whereas it was found to be 12% without GNPs. When the air from the operator’s mouth was analyzed just after rinsing with an antiseptic mouthwash, the ethanol content was very high (3.5 *v*/*v*%). The background level was reached only after rinsing with water.

## 1. Introduction

The mouth, due to its role as an interface between the outside world and the interior of the body receiving food, is exposed to multiple potentially pathogenic agents. The active ingredients in antiseptic mouthwashes must be effective against bacteria and fungi, particularly of the *Candida* type. Their antifungal action is indeed required to prevent oral candidiasis. These antiseptics are either chlorhexidine, cetylpyridinium chloride, hexetidine, hydrogen peroxide, sodium benzoate, or povidone iodine. The main role of ethanol is to solubilize substances responsible for flavor, or active molecules, to increase their bioavailability. Following the positive alcohol test—in the workplace, by the employer—of an employee who denied any consumption, the occupational health service was asked about the possible interference of a mouthwash containing alcohol taken in the minutes preceding breath alcohol measurements. The permitted level is 0.25 mg/L of ethanol in the exhaled air (0.012 *v*/*v*% or 120 ppm), which should correspond to 0.5 g/L of ethanol in blood in case of ingestion of alcohol [1]. If the test is positive, this can have economic and socio-professional consequences for employees. The data available in the literature support the hypothesis that a product containing alcohol used as a mouthwash may interfere with alcohol measurements in the expired air in the first 15 min [1].

Numerous sensors for the detection of ethanol are based on semiconducting metal oxides, as presented in a recent review by Weimar et al. [2]. Gardner et al. showed that the sensor based on the electrical resistance of ZnO nanorods, working at 350 °C, is influenced by humidity; the ethanol on acetone signal ratio was found to be 3.3 [3]. A ZnO nanoparticle network, obtained via in situ annealing of a porous metal-organic framework (MOF), ZIF-8, working at 300 °C, gave a resistive signal for ethanol and acetone with a ratio of 1.87 [4]. Gardner et al. deposited an Au-SnO_2_ nanocomposite on a CMOS-MEMS platform for the detection of ethanol at 400 °C, and the resistive signal for ethanol was 3.5 times higher than that of acetone [5]. Under visible light illumination, Di Natale et al. studied the porphyrin-functionalized ZnO nanorod photoconductivity changes, modulated by exposure to ethanol and trimethylamine. The signal for trimethylamine was found to be 150 times higher than that of ethanol [6]. Other sensors for the detection of ethanol are based on carbonaceous nanomaterials. Using a carbon black/polyvinylpyrrolidone composite chemoresistor, working between 25 °C and 55 °C, Gardner et al. detected ethanol with a quantification limit of 2270 ppm and an ethanol on methanol signal ratio of 0.75 [7]. Wilson et al. presented a resistor sensor based on reduced graphene oxide (rGO), working at room temperature. The same sensitivity was obtained for ethanol and methanol, this sensitivity being influenced by humidity [8]. Many sensors based on carbon nanotubes (CNTs) were able to detect ethanol. Recently, Shaalan et al. [9] obtained, with CNTs grown on 24 nm of Ni, a quantification limit of 5 ppm for ethanol and an ethanol on acetone signal ratio of 5.2.

Taking advantage of their higher selectivity, compared to that of these nanomaterials, enzyme-based sensors were used as ‘‘bio-sniffers’’ for the detection of ethanol, working at ambient temperature. The enzyme alcohol dehydrogenase (ADH) can be used for this purpose. It is necessary to add its cofactor, the oxidized form of nicotinamide adenine dinucleotide (NAD^+^). The enzymatic reaction, leading to the oxidation of ethanol into acetaldehyde and the release of one proton per oxidized ethanol molecule, is as follows:$${\mathrm{CH}}_{3}{\mathrm{CH}}_{2}{\mathrm{OH}+\mathrm{NAD}}^{+}\overset{\mathrm{ADH}}{\to }{\mathrm{CH}}_{3}{\mathrm{CHO}+\mathrm{NADH}+H}^{+}$$

Different transduction modes can be used: amperometry [10,11,12] by the detection of NADH, capacitance [13], conductometry by the detection of the release of protons [14], or fluorimetry by detecting the autofluorescence of the coenzyme NADH [15,16,17,18]. As the sensors are miniaturized, the enzyme must be encapsulated in a hydrophilic film in order to keep it active. Alcohol dehydrogenase was encapsulated in a mixture of DEAE-dextran and hydroxyethyl cellulose, in an amperometric biosensor [10]. PAMAM/ADH layer-by-layer films deposited onto interdigitated electrodes (IDEs) were used for capacitance measurements [13]. An ADH-NAD^+^ chitosan composite was electrodeposited on IDEs for conductometric measurements [11]. An enzyme-based fluorometric electrospun fiber sensor (eFES) mesh including ADH and NAD^+^ in a polyvinyl alcohol polymeric matrix was used for fluorometric measurements of ethanol vapor [18].

Chitosan is a polysaccharide extracted from shrimp shells after deacetylation of chitin; it is biocompatible and has been shown to increase the storage stability of laccase and to improve its antibacterial activity [19]. It can be electrochemically deposited on a polarized cathode, due to its local insolubility in the high pH value at the electrode surface through the electroreduction of protons [20]. This matrix has already been used for the immobilization of ADH and NAD^+^ on IDEs for the conductometric detection of ethanol [14]. The advantages of this conductometric sensor, based on IDEs, are its miniaturization, and its use in a differential mode between a working sensor and a reference sensor which allows the cancellation of any drift. The detection limit obtained with this enzymatic sensor was rather high, 1200 ppm of ethanol. The sensitivity of a conductometric urea sensor, working in a liquid medium, was shown to increase by a factor of 10 by including gold nanoparticles (GNP) in the vicinity of the enzyme urease [21]. In Reference [22], in which the spatial variation of conductivity is modelled, it was found to be higher at the transducer surface. Due to the conductive properties of gold nanoparticles, they can behave as microelectrodes; in their presence, the new electrical field lines will then be closer to the transducer surface [23].

In this present work, a micro conductometric ADH-NAD^+^-GNP chitosan composite-based ethanol sensor was designed. Its analytical performance was obtained by measurements in the headspace over aqueous solutions of ethanol and other solvents (methanol, acetone). The ethanol sensor was then used for the detection of ethanol in a commercial mouthwash and for the detection of ethanol vapor in the operator’s mouth after using a mouthwash and after rinsing his mouth with water. It was then possible to give recommendations for the use of mouthwashes with regard to the rules about alcohol consumption in the workplace.

## 2. Materials and Methods

### 2.1. Reagents

Alcohol dehydrogenase from *Saccharomyces cerevisiae* (300 units/mg), nicotinamide adenine nucleotide from yeast (>98%), bovine serum albumin (BSA) (96%), chitosan (deacetylated chitin, degree of deacetylation 80.0–95%, molecular weight Mw = 250 kDa), acetic acid (99.9%), 0.1 M sodium hydroxide solution, potassium chloride (99.0%), phosphate buffer saline (PBS) tablets, Gold (III) chloride trihydrate (≥99.999%), trisodium citrate dihydrate, sodium chloride (NaCl), ethanol (99.5%), methanol (99.5%), and acetone (99.5%) were purchased from Sigma. Hydrochloric acid (37%) and nitric acid (68%) were purchased from BDH Prolabo-VWR International.

Ultra-pure water (UPW) (resistivity > 18 MOhm.com) was produced by a Millipore System (Merck, Darmstadt, Germany).

### 2.2. Microconductometric Chip

A micro conductometric chip is presented in Figure 1. It was fabricated in CNM-CSIC, Barcelona, Spain. The silicon chip size was 7300 × 4100 μm. Devices with a gold finger width of 60 µm and finger separation of 60 µm were designed. The dimensions of the gold IDE areas are a diameter of 1200 μm for the circular devices, and dimensions 1740 μm × 900 μm for the central rectangular one. The flowchart of the process of fabrication of the microconductometric chip is presented in Ref. [14]. The microconductometric chip was bonded to a standard printed circuit board (PCB) using wire-bonding (aluminum wire, F 25 μm) by a Kulicke and Soffa 4523A Digital Instrument (Eindhoven, The Netherlands) and then the interconnects were protected with epoxy resin, EPOTEK H70E2LC from Epoxy Technology (Orgeval, France).

### 2.3. Synthesis of the Gold Nanoparticles (GNPs)

The gold nanoparticles were synthesized by citrate reduction of HAuCl_4_·3H_2_O with minor modifications, as described previously [24]. In brief, a solution of HAuCl_4_·3H_2_O 1mM was mixed with deionized water in the ratio of 1:1 in a 250 mL flask and stirred and heated until it boiled. Then, the reducing agent solution (38.8 mM trisodium citrate dihydrate) was immediately added to the solution. The stirring process continued for 20 min until the solution color changed from pale yellow through dark red to wine-red. The magnetic bars and glassware were washed with aqua regia (HCl: HNO_3_ 3:1, *v*/*v*) before being rinsed with water, which prevented GNP aggregation from the previous residual and prevented unwanted nucleation in the step of the synthesis.

### 2.4. Fabrication of the Micro Conductometric Ethanol Sensor

The flowchart of the fabrication of the ethanol sensor is presented in Figure 2.

A chitosan suspension (0.1g/mL) in 0.1 M acetic acid was prepared, pH value being adjusted to 7 by the addition of some drops of 0.1 M NaOH solution. This suspension was agitated for two days to homogenize it, and then refrigerated overnight, until the pH stabilized.

The working sensor (ADH-NAD^+^-GNP-chitosan composite) was prepared by adding 2.5 mg of ADH, 0.22 mg of NAD^+^ [9:1), and 500 µL of GNPs suspension (concentration: 2.8 × 10^12^ GNPs/mL) to 2.5 mL of chitosan suspension.

The reference sensor (BSA-GNP-chitosan composite) was prepared by adding 2.7 mg of BSA and 500 µL of GNPs suspension (concentration: 2.8 × 10^12^ GNPs/mL) to 2.5 mL of chitosan suspension.

Before electrodeposition, the bonded and encapsulated microductometric chips were cleaned first with ethanol under ultrasonication for 15 min, then with acetone, then rinsed with UPW, dried under a nitrogen flow, and then exposed to a UV–Ozone ProCleaner (BioForce, Gifu, Japan) for 30 min. Electrodeposition was carried out with a three-electrode format, the microconductometric chip as the working electrode (the connected four circular devices), an Ag/AgCl electrode as the reference electrode, and a platinum wire counter electrode. Chronoamperometry was carried out at an applied potential of −1.4 V for 3 min using a VMP3 multichannel potentiostat (Biologic EC-Lab, Seyssinet-Pariset, France).

### 2.5. Micro Conductometric Measurements

Conductometric detection was achieved by applying to each pair of IDEs (sensors) a small-amplitude sinusoidal voltage (10 mV peak-to-peak at 0 V) at a 10 kHz frequency generated by a “VigiZMeter” conductometer manufactured by the company “Covarians”, and the responses of the gas sensor were recorded at room temperature (23 ± 1 °C), as a function of time. The differential output signal was recorded between the working and the reference pairs of IDEs. The working sensor was obtained by electro-deposition of an ADH-NAD^+^-GNP-chitosan composite on the four circular devices, and the reference sensor was obtained by the electro-deposition of a BSA-GNP-chitosan composite on the four circular devices. Conductometric measurements were made after introducing the working sensor and the reference sensor in the headspace (volume of the gas exposure chamber: 15 cm^3^) over the liquid phase (volume: 10 cm^3^) in a cylindrical container for one minute and then withdrawing it (Figure 3). The differential measurement of conductance (ΔG) was recorded versus time. The response time (**t_Res_**) describes the time necessary to reach 90% of the total change of conductance and the recovery time (**t_Rec_**) characterizes the time necessary to recover 10% of the total change in conductance, as defined in Ref. [25].

The ethanol sensor performance was tested in the headspace above aqueous solutions of ethanol, methanol, and acetone with known concentrations of between 0–100%. The as-mentioned gas phase concentration depends on Henry’s law constants of the given analyte in water at 25 °C and was calculated from Henry’s law following the equation formulated by Sander in 1999 [26].
(1)$$k_{H}^{0}=\frac{c_{a}}{p_{g}}$$
where $k_{H}^{0}$ is Henry’s constant in standard conditions; [$k_{H}^{0}$] = M/atm, $c_{a}$ is the aqueous concentration of the analyte; [ca] = M and $p_{g}$ is the partial pressure of the analyte in the gas phase; and [*p_g_*] = atm.

For the gas analytes listed above (at 25 °C), i.e., methanol, ethanol, and acetone, Henry’s constant values of 2.2 × 10^2^ M/atm, 1.9 × 10^2^ M/atm, and 0.24 × 10^2^ M/atm respectively were considered [27]. In Table 1, the calculated equilibrium gas-phase concentrations of methanol, ethanol, and acetone above the aqueous phase are listed.

### 2.6. Characterisation Techniques

SEM images were obtained using a Tescan Vega SBU scanning electron microscope (Tescan group, Brno, Czech Republic), operating at an accelerating voltage of 20 kV, equipped with a Bruker Esprit Compact EDS detector (Bruker, Kontich, Belgium).

Transmission electron microscopy (TEM) was performed with a JEM-1400 flash microscope from JEOL (Tokyo, Japan) at the Centre Technologique des Microstructures, University Claude Bernard Lyon1. The TEM image could be used to observe the morphology of the particles. A drop of the sample was deposited onto a carbon grid and left until it dried. ImageJ software 8B calculated the gold nanoparticle size and polydispersity index (PDI) from TEM images of at least 200 particles. The absorbance of the gold nanoparticles was scanned with a range of 300–800 nm by a Jenway 7250 UV/Visible Spectrophotometer (Jenway, London, UK).

FTIR spectroscopy was run at room temperature using a continuous Nicolet vacuum microscope (Thermo Fisher Scientific, Waltham, MA, USA) coupled with Nexus infrared spectroscopy (Nexus Analytics, Singapore) in specular reflectance mode with an MCT detector (Infrared Associates Inc., Stuart, FL, USA). The recordings were obtained with a resolution of 4 cm^−1^, a spectral width between 690 and 4000 cm^−1^, and signal processing was carried out by Happ–Genzel apodization (256 scans).

## 3. Results and Discussion

### 3.1. Preparation of the Ethanol Microsensor

#### 3.1.1. Characterization of the GNPs

TEM image of the synthesized gold nanoparticles showed spherical monodispersed particles in size and shape (Figure 4A). Gold nanoparticle average size and their polydispersity index (n = 210) presented at 15.21 ± 1.90 nm (Figure 4B). The gold nanoparticle solution color was wine-red, which revealed the characteristic surface plasmon bands. They were present in the intense absorption spectrum where the maximum absorbance peak was 519 nm (Figure 4C).

#### 3.1.2. Electrodeposition of the Chitosan Composite Film

The cathodic current at the electrode, measured as a function of time, fluctuated according to the diffusion of H^+^ from the bulk solution towards the sensor surface. The current (mA) continued to decrease from 40 µA until it became stable within 4 min and equal to 5 µA, which means that the chitosan composite film is well deposited on the gold surface to form a film. The film was deposited on the four circular devices as shown in Figure 5A. The gold nanoparticles were dispersed in the chitosan composite film as shown in the Au EDS cartography above and between the gold IDEs (Figure 5B). The FTIR spectra of pure chitosan film and of a chitosan composite film are presented in Figure 6. From the FTIR spectra of chitosan (Figure 6, red curve), C–N at 1151 cm^−1^, N–H at 1574 cm^−1^, C–O at 1071 cm^−1^, and C–C at 1256 cm^−1^ are present [28]. In the chitosan film including NAD^+^ (Figure 6, blue curve), P=O symmetric stretch at 1032 cm^−1^, ribose moiety and P–O stretch at 1110 cm^−1^, P=O asymmetric stretch at 1204 cm^−1^, C=N stretch at 1342 cm^−1^, nicotinamide and adenine moieties at 1475 cm^−1^ are present [29]. Stretching of amide I at 1657 cm^−1^ and stretching of amide II at 1575 cm^−1^ originating from chitosan and from ADH are present.

### 3.2. Effect of the Presence of GNP in the Chitosan Composite Film on the Conductometric Measurements of Ethanol Vapor

In Figure 7, the response of a microconductometric sensor with GNPs inserted in the chitosan composite film and that of a microconductometric sensor without GNPs, in a differential measuring mode, in the headspace above a pure ethanol solution, can be compared. The response in the presence of GNPs is 4000 µS/cm and the response in the absence of GNPs is 390 µS/cm. This demonstrates the effect of GNPs on the detection of ethanol vapor very well. This result confirms what has already been observed with a urease-based sensor working in liquid phase [21].

The response of the micro conductometric sensor with GNPs, in a differential measuring mode, in the headspace above different pure solvents (ethanol, methanol, acetone, toluene, chloroform and water) is presented in Figure 8. A large signal for ethanol is obtained compared to the other solvents, even methanol and acetone. The response for the non-polar solvents (toluene and chloroform) is almost zero. The response to pure water is 2% of the response to ethanol. This response is due to the hydrophilicity of the chitosan composite film, and it is reduced due to the differential measuring mode.

### 3.3. Analytical Performance of the Ethanol Sensor

The response time of the ethanol sensor, in the headspace over the pure ethanol solution is 10 s and the recovery time is 5 s (Figure 9). For an ethanol sensor with a chitosan composite film without GNPs ([10], the response time was 21 s and the recovery time was 58 s, showing higher reversibility in the presence of GNPs. This phenomenon could be due to higher porosity in the presence of GNPs.

The conductometric signal of the ethanol sensor was measured in the headspace of chloroform solutions of different volumetric percentages of ethanol (100%, 80%, 60%, 40%, 20, 10%, 5%, 2%, 1%), respectively, corresponding to the different *v*/*v*% in the gaseous phase (9, 7.2, 5.4, 3.6, 1.8, 0.9, 0.45, 0.18, 0.09) (Figure 10A,B). The corresponding calibration curve is presented in Figure 10C. The measured sensitivity for the ethanol sensor is 416 µS/cm (*v*/*v*%)^−1^ for ethanol vapor. The detection limit for ethanol is 106 ppm. Without GNP, the micro conductometric ethanol sensor’s sensitivity was 36.8 µS/cm (*v*/*v*%)^−1^ [14], so enhanced by a factor of 11.3. This phenomenon has already been observed for a urea sensor (enhancement factor 10) [21] and for an acetylcholine sensor (enhancement factor: 2.3) [30].

The relative standard deviation (RSD) obtained with the same sensor is 2%. It is noticeable that without GNPs, the ethanol sensor’s RSD is 12%. The improvement of the RSD was also observed for an acetylcholine sensor, a butyrylcholine sensor, and a glucose sensor [25]. The sensor retains its detection sensitivity for one month, when kept in a fridge at 4 °C between measurements. The inter-sensor reproducibility obtained for five sensors was 8%.

The measured sensitivity is 50 µS/cm (*v*/*v*%)^−1^ for methanol vapor and 2.6 µS/cm (*v*/*v*%)^−1^ for acetone vapor. This sensor is 8.3 times more sensitive to ethanol than to methanol, and 160 times more sensitive to ethanol than to acetone. The presence of GNPs was found to improve the selectivity factor. Without GNPs, the selectivity factor to methanol was found to be lower (2.6) than it was to acetone (28.3) [14].

A comparison of the response times and detection limits of the micro conductometric ADH-based ethanol sensor with those of previously published enzymatic ethanol sensors (Table 2). Response time of the fabricated ethanol sensor is in the low range compared to the others, whereas the detection limit is still in the high range.

### 3.4. Effect of Mouthwash on the Content of Ethanol in the Exhaled Air

The ethanol content in the mouthwash (Eludrilpro from Pierre Fabre, Paris, France) was detected with the ethanol sensor in the headspace of the mouthwash. The conductometric measurements are presented in Figure 11. In the headspace of the mouthwash, the signal was 1632 µS/cm, which corresponds to 42.0 ± 0.8 *v*/*v*% in the mouthwash. The declared content by the provider was 41 *v*/*v*%.

The effect of the use of mouthwash was then studied. The ethanol sensor (with the reference sensor) was introduced into the operator’s mouth after rinsing with mouthwash. The diagram of the experimental set-up is presented in Figure 2B.

As shown in Figure 11B, the micro conductometric signal, just after rinsing with the mouthwash, is 1500 µS/com, corresponding to a concentration of 3.5 *v*/*v*% of ethanol in the air of the operator’s mouth, 291 times higher than the allowed level in cases of alcohol ingestion. After waiting for 3 min and rinsing with water, the signal level returns to the level before the mouthwash, due to the humidity in the exhaled air. This result is in agreement with what was observed in Ref [1], using a classical alcoholmeter used by the police. This result validates this ethanol sensor that can be convenient for personal use.

## 4. Conclusions

This work shows how the addition of gold nanoparticles can greatly improve the analytical performance of an ADH-based micro conductometric sensor: decrease of the response time by a factor of 2.1, increase of the sensitivity by a factor of 11.3, decrease of the detection limit up to 106 ppm, relative standard deviation decreased by a factor of 6, and a higher selectivity factor (multiplied by 3.2) to methanol. The comparison of the selectivity of detection of the present ethanol sensor to that of the ethanol sensors based on ZnO, carbonaceous nanomaterials, shows the interest of the fabricated ADH-based sensor with an ethanol signal 8.3 times higher than that of methanol and 160 times higher than that of acetone. The obtained ethanol sensor was used for testing the effect of mouthwash on the ethanol content of exhaled breath. This point is of importance for tests on workers in workplaces, and on drivers. The ethanol sensor showed a high level of ethanol in the operator’s mouth just after rinsing with the commercial mouthwash; only after rinsing with water does the sensor signal reach the background signal (less than 106 ppm). The prepared ethanol sensor is thus of interest for personal use due to its easy use.

## Figures and Tables

**Figure 1 nanomaterials-13-02316-f001:**
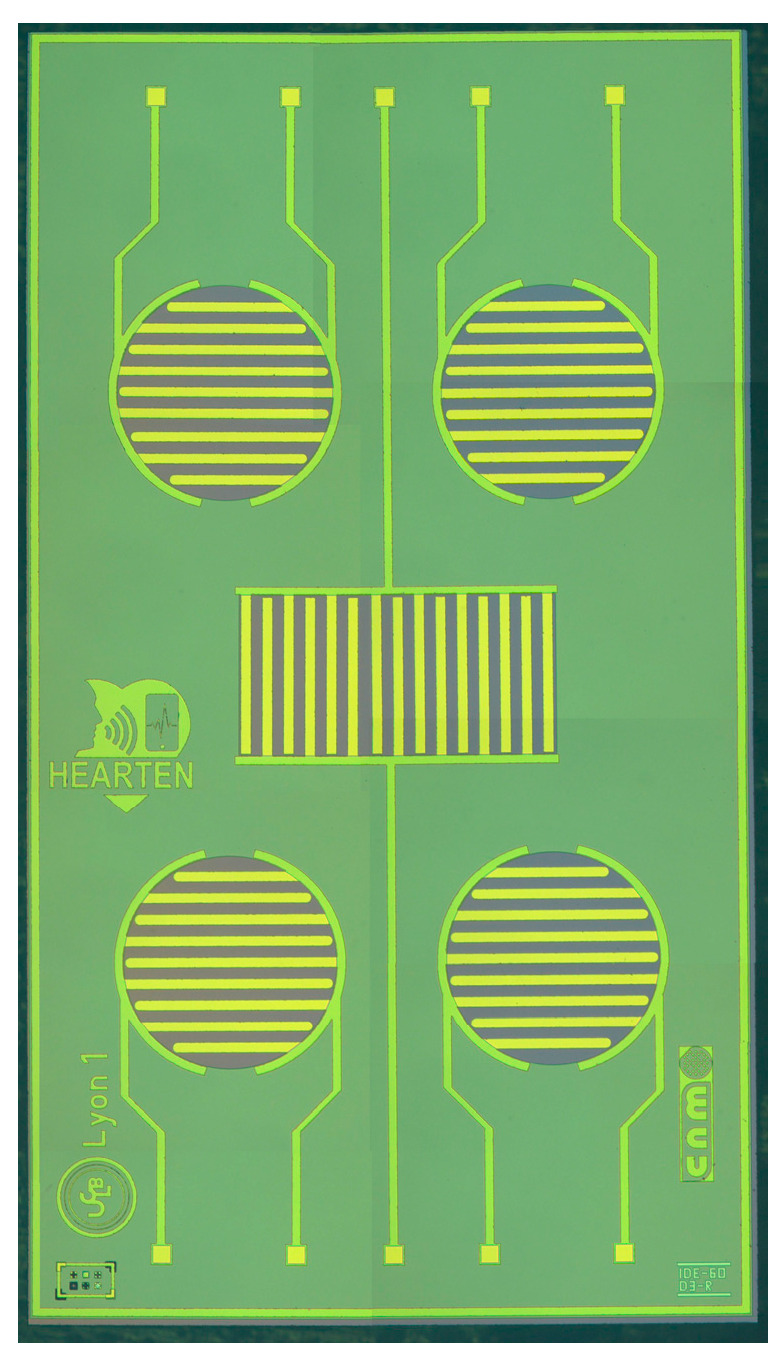
Photo of the micro conductometric chip (8× magnification).

**Figure 2 nanomaterials-13-02316-f002:**
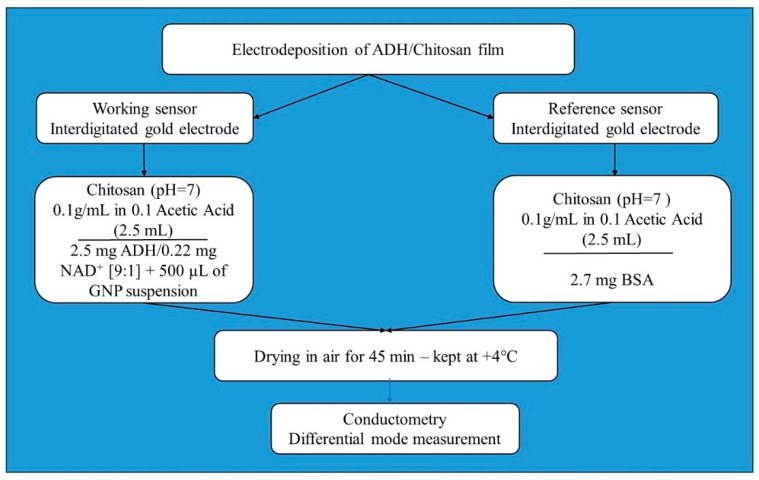
Flowchart of the fabrication of the micro conductometric ethanol sensor.

**Figure 3 nanomaterials-13-02316-f003:**
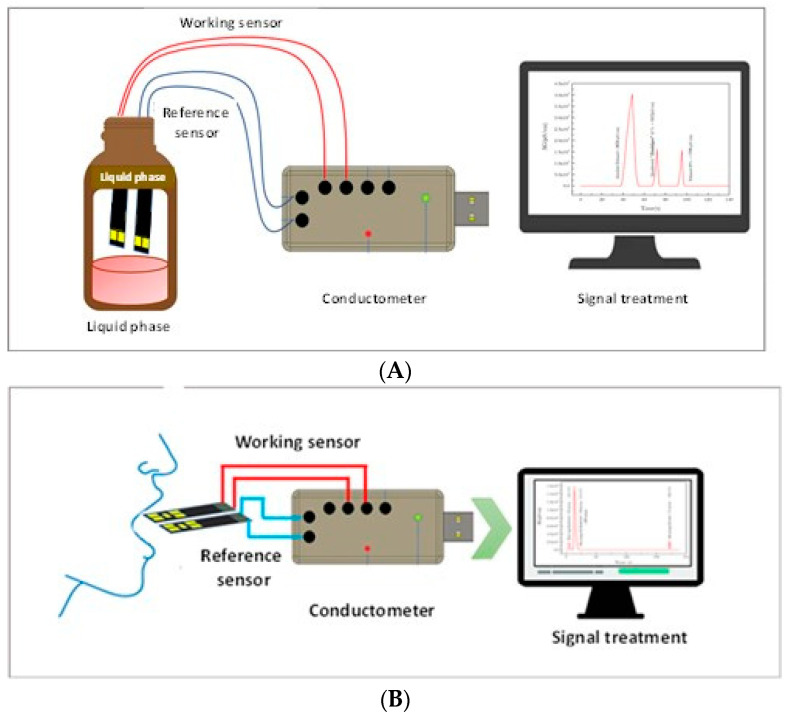
Diagram of the experimental setup. (**A**) Measurement in headspace above the liquid phase. (**B**) Monitoring of ethanol in the operator’s mouth.

**Figure 4 nanomaterials-13-02316-f004:**
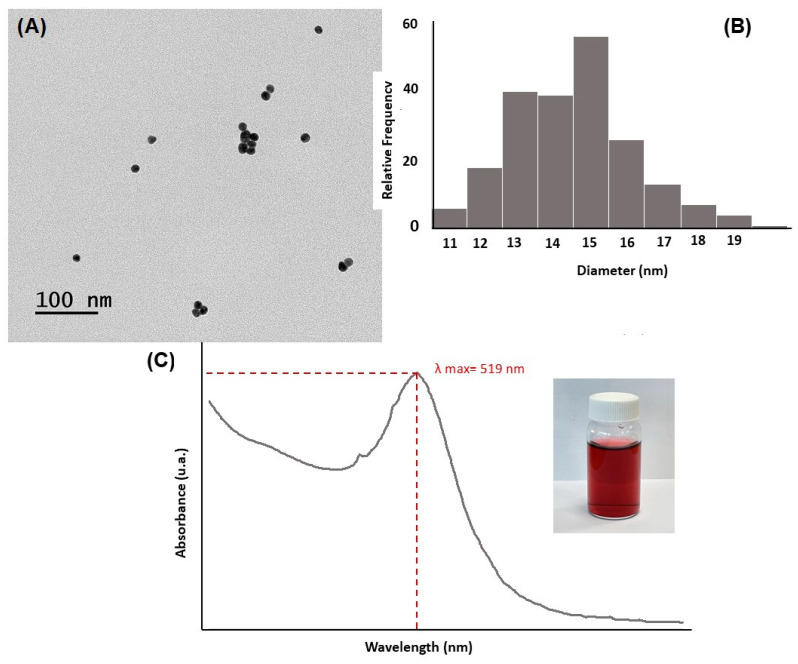
Characterization of synthesized gold nanoparticles: (**A**) TEM image of the nanoparticles, scale bar 100 nm, (**B**) particle dimensional dispersion histogram, and (**C**) UV-VIS spectra.

**Figure 5 nanomaterials-13-02316-f005:**
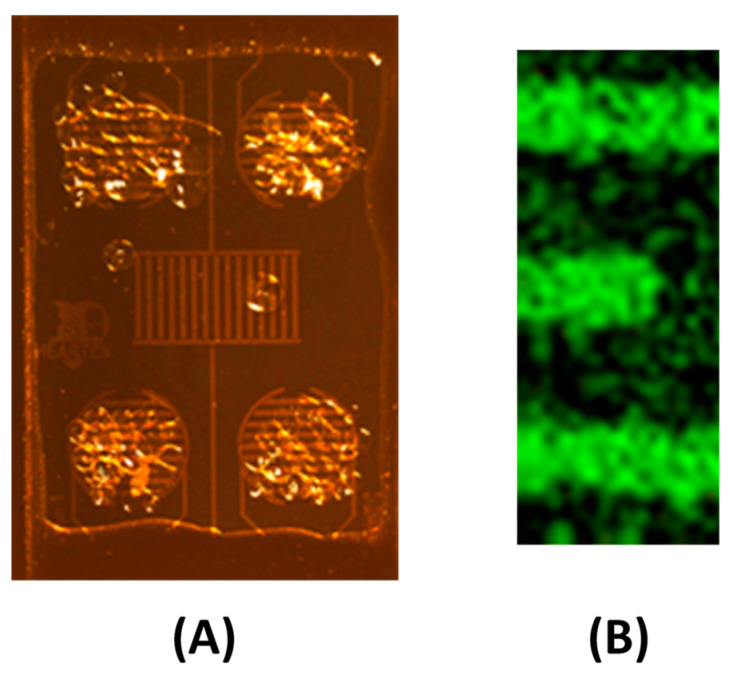
(**A**) Electrodeposited chitosan composite film on the four circular devices of the microconductometric chip. (**B**) EDS cartography of gold showing the dispersion of GNPs in the chitosan composite film.

**Figure 6 nanomaterials-13-02316-f006:**
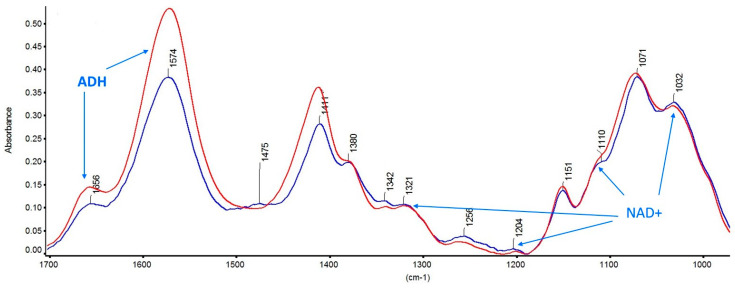
FTIR spectra of pure chitosan film (red curve) and chitosan film including ADH and NAD+ (blue curve).

**Figure 7 nanomaterials-13-02316-f007:**
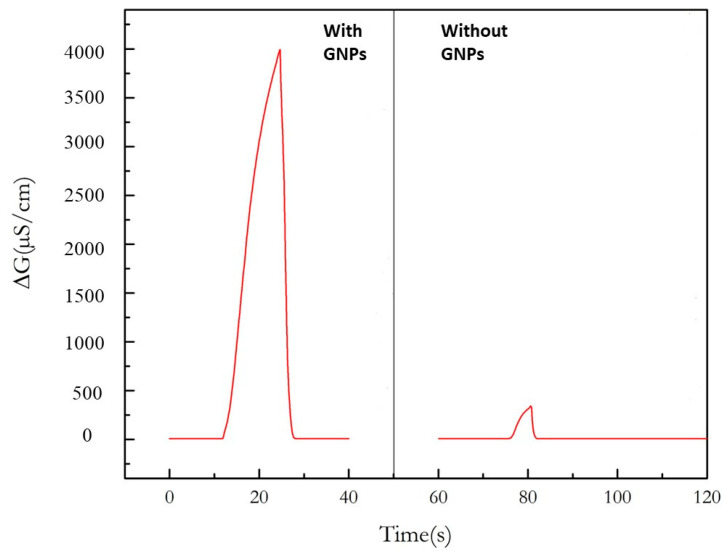
Response of a micro conductometric sensor with GNPs inserted in the chitosan composite film and response of a micro conductometric sensor without GNPs, in a differential measuring mode, in the headspace above a pure ethanol solution.

**Figure 8 nanomaterials-13-02316-f008:**
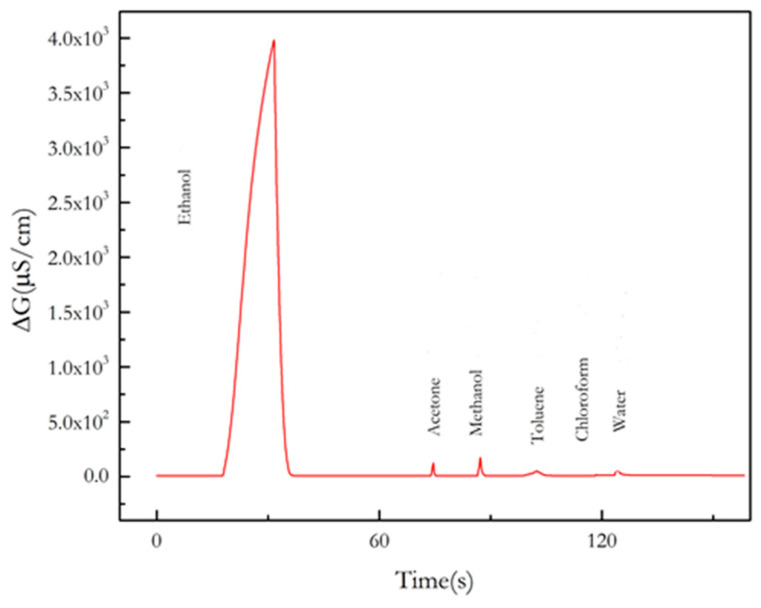
Response of the micro conductometric sensor with GNPs, in a differential measuring mode, in the headspace above different pure solvents (ethanol, methanol, acetone, toluene, chloroform, and water).

**Figure 9 nanomaterials-13-02316-f009:**
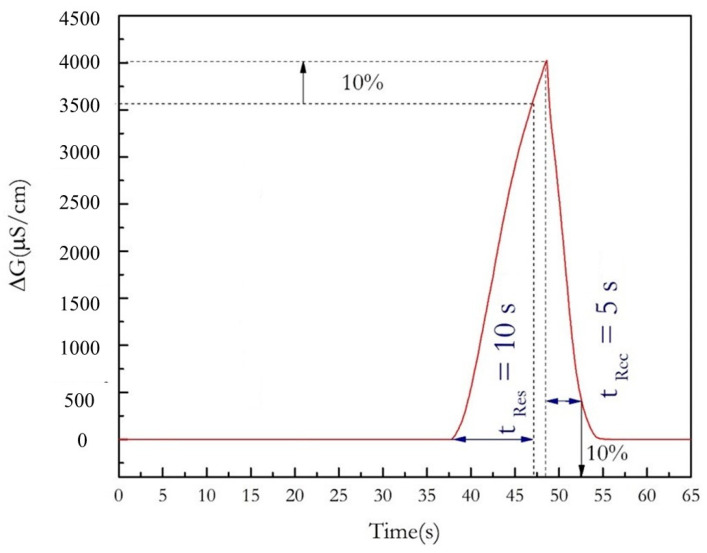
Measurements of the response time (tRes) and of the recovery time (tRec) on the real-time registration of the ethanol sensor response (liquid phase: pure ethanol).

**Figure 10 nanomaterials-13-02316-f010:**
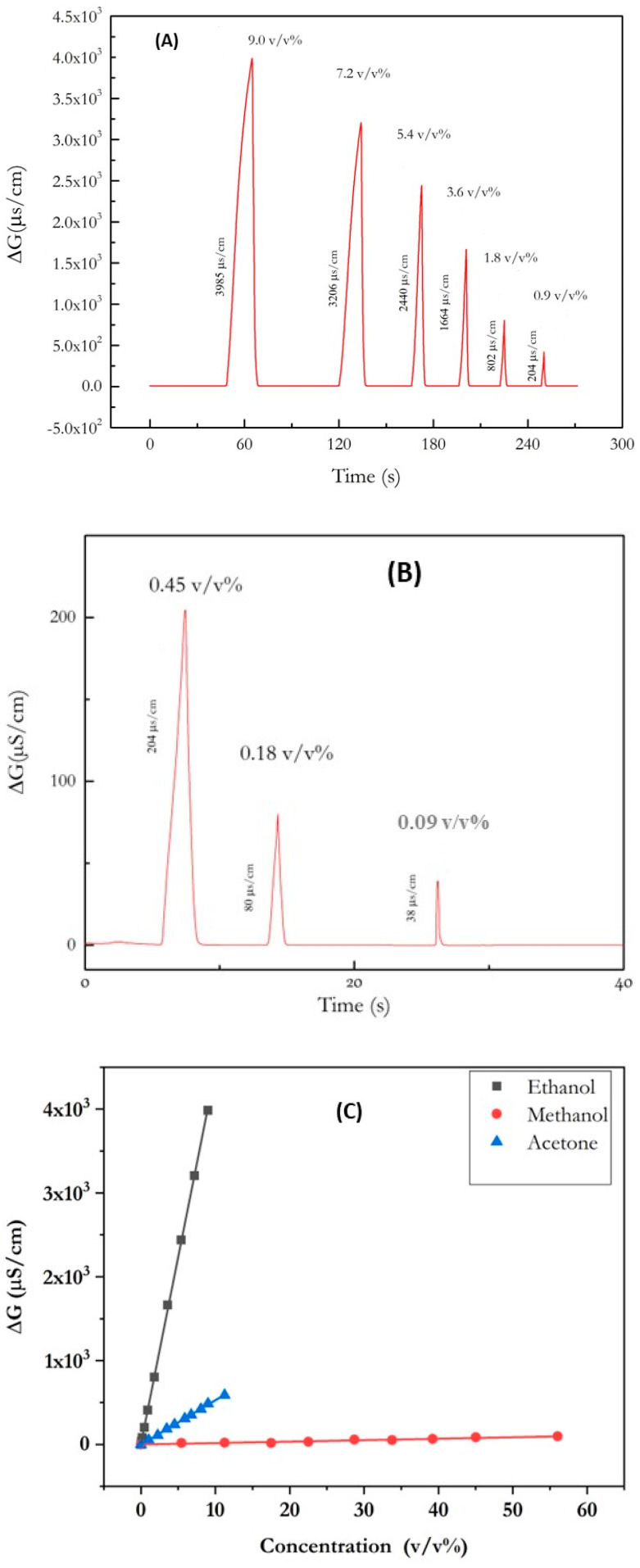
(**A**) Micro conductometric response of the ethanol sensor in the headspace of chloroform solutions of different volumetric percentages of ethanol (9.0 *v*/*v*%, 7.2 *v*/*v*%, 5.4 *v*/*v*%, 3.6 *v*/*v*%, 1.8 *v*/*v*%, and 0.9 *v*/*v*%). (**B**) Micro conductometric response of the ethanol sensor in the headspace of chloroform solutions of different volumetric percentages of ethanol (0.45 *v*/*v*%, 0.18 *v*/*v*%, 0.09 *v*/*v*%) (**C**) Calibration curves of the ethanol sensor in the headspace of aqueous solutions of different volumetric percentages of ethanol, methanol, and acetone.

**Figure 11 nanomaterials-13-02316-f011:**
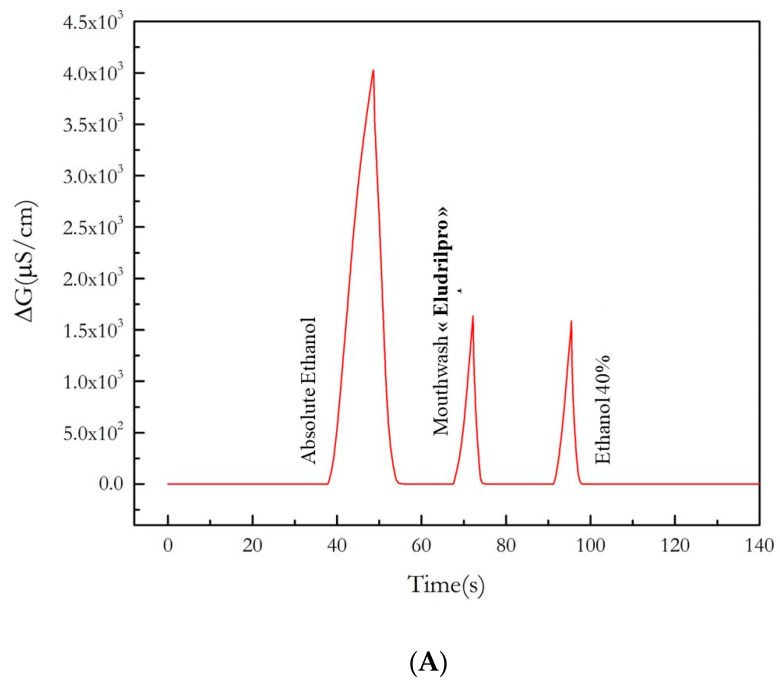
(**A**) Conductometric signal in the headspace of the mouthwash, of pure ethanol and of 40 *v*/*v*% aqueous ethanol solution. (**B**) Effect of rinsing of the operator’s mouth with a mouthwash: conductometric signal of the ethanol sensor, before rinsing, after rinsing, and after rinsing with water on the air content in the operator’s mouth.

**Table 1 nanomaterials-13-02316-t001:** Equilibrium gaseous phase concentrations above aqueous ethanol, methanol, and acetone solution at 25 °C calculated through the equations reported by Sander et al. [26].

Volumetric Percentage in the Liquid Phase	Volumetric Percentage of Ethanol in the Gaseous Phase *v*/*v*%	Volumetric Percentage of Methanol in the Gaseous Phase *v*/*v*%	Volumetric Percentage of Acetone in the Gaseous Phase *v*/*v*%
0%	0	0	0
20%	1.79	2.25	11.26
40%	3.58	4.5	22.52
60%	5.37	6.75	33.78
80%	7.16	9	45.04
100%	8.95	11.25	56.03

**Table 2 nanomaterials-13-02316-t002:** Response times and detection limits of previously published enzymatic ethanol sensors.

Type of Sensor	Response Time (T_Res_)	Detection Limit	Ref.
ADH/amperometric	5 s	20 ppm	[10]
AO/amperometric	69 s	0.5 ppm	[11]
ADH/fluorometric	20 s	0.5 ppm	[15]
ADH/fluorometric	20 s	0.1 ppm	[16]
ADH/conductometric	21s	1200 ppm	[14]
ADH/conductometric	10 s	106 ppm	This work

## Data Availability

Data will be made available on request.

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
