# Peer review of "A Sensitive Micro Conductometric Ethanol Sensor Based on an Alcohol Dehydrogenase-Gold Nanoparticle Chitosan Composite"

_nanomaterials, 2023, doi:10.3390/nano13162316_

Round 1

Reviewer 1 Report

The Authors in their manuscript entitled Study of the effect of mouthwash on the ethanol content of exhaled air using a micro conductometric sensor based on an alcohol dehydrogenase-gold nanoparticle chitosan composite” proposed alternative instrument for ethanol sensing with possible utility after rinsing of mouth. The Authors seems to be familiar with sensors technology, therefore the manuscript is written clearly. However, some minor aspects were overlooked, or too superficially treated in this article. Therefore, I recommend to revise the manuscript before submission to Nanomaterials.

Major and minor critique points:

1.      The authors should prepare graphical abstract to make the manuscript more attractive for potential readers.

2.      The abstract is written in a very general way, the details about the analytical parameters should be added.

3.      The Authors should revised manuscript according to Guide for authors attached in journal (e.g. Tables, figures captions, etc.). Also, all Latin phrases (via, i.e., e.g., in-situ, et al., etc…) in scientific writing should be in italics and abbreviations should be explained while using for the first time. Moreover, scientific writing rules concerning units spacing should be included: https://physics.nist.gov/cuu/Units/checklist.html and journal internal instructions. Moreover, to be honest, the manuscript is full of colloquialisms, mistakes, errors and typos. Some of the fundamentals are presented as mental shortcuts. In some cases sentences are hard to understand because of this.

4.      The Introduction section is lack of current publications- original, review, opinion articles. Reference section needs huge refreshment to be trendier and more comprehensive for potential readers. Refs published by specialists in the sensors technology need to be included. The specialists like: Wilson, Gardner, Gebicki, Di Natale, et al. should be included in the Introduction section. Their articles from last 5 years are the best examples of recent trends in the field of sensors and biosensors.

5.      Figure with the scheme of sensors’ construction stages should be included.

6.      Some Figures should be merged, e.g. Fig 2 and 11 and some Figs with the results.

7.      What flow conditions (sccm) were used during the sensing experiments? What were the dimensions (size, etc.) of the gas sensing chamber? These parameters can have a significant impact on the sensor signal and were not discussed at all. Samples were heated somehow or measurements were made in STP?

8.      Sensors drift were evaluated and somehow?

9.      The conclusion section of the article presents the superiority of the proposed method. But highlight the important outcomes obtained from this research which establish that the methodology is better than the previous methods. However, novelty statement of presented studies should be clearly presented also perspectives should be outlined.

Expert language editing is obligatory, it should greatly improve the manuscript. It should be proofread and edited to improve the English throughout understand.

Author Response

Manuscript ID: nanomaterials-2513820

Title: A sensitive micro conductometric ethanol sensor based on an alcohol dehydrogenase-gold nanoparticle chitosan composite.

Responses to reviewer 1’ comments

The authors thank the reviewer for valuable comments that will improve the quality of the manuscript.

Reviewer 1

The Authors in their manuscript entitled “Study of the effect of mouthwash on the ethanol content of exhaled air using a micro conductometric sensor based on an alcohol dehydrogenase-gold nanoparticle chitosan composite” proposed alternative instrument for ethanol sensing with possible utility after rinsing of mouth. The Authors seems to be familiar with sensors technology, therefore the manuscript is written clearly. However, some minor aspects were overlooked, or too superficially treated in this article. Therefore, I recommend to revise the manuscript before submission to Nanomaterials.

Major and minor critique points:

  1. The authors should prepare graphical abstract to make the manuscript more attractive for potential readers.

A new graphical abstract was prepared

  1. The abstract is written in a very general way, the details about the analytical parameters should be added.

      The abstract was rewritten and the details about the analytical parameters obtained in presence of gold nanoparticles were added: The response time is reduced to 10 s, compared to 21 s without GNPs. The sensitivity is 416 µS/cm (v/v%)-1 which is 11.3 times higher than without GNPs. The selectivity factor versus methanol is 8.3, three times higher than without GNPs. The relative standard deviation (RSD) obtained with the same sensor is 2%, whereas it was found to be 12% without GNPs.»

  1. The Authors should revised manuscript according to Guide for authors attached in journal (g. Tables, figures captions, etc.). Also, all Latin phrases (via, i.e., e.g., in-situ, et al., etc…)in scientific writing should be in italics and abbreviations should be explained while using for the first time. Moreover, scientific writing rules concerning units spacing should be included: https://physics.nist.gov/cuu/Units/checklist.html and journal internal instructions. Moreover, to be honest, the manuscript is full of colloquialisms, mistakes, errors and typos. Some of the fundamentals are presented as mental shortcuts. In some cases sentences are hard to understand because of this.

The manuscript was totally revised by a native English person.

  1. The Introduction section is lack of current publications- original, review, opinion articles. Reference section needs huge refreshment to be trendier and more comprehensive for potential readers. Refs published by specialists in the sensors technology need to be included. The specialists like: Wilson, Gardner, Gebicki, Di Natale, et al. should be included in the Introduction section. Their articles from last 5 years are the best examples of recent trends in the field of sensors and biosensors.

The cited references about ethanol sensors were modified according to the reviewer’s suggestions: “Numerous sensors for the detection of ethanol are based on semiconducting metal oxides, as presented in a recent review by Weimar et al [2]. Gardner et al showed that the sensor based on the electrical resistance of ZnO nanorods, working at 350°C, is influenced by humidity; the ethanol on acetone signal ratio was found to be 3.3 [3]. A ZnO nanoparticle network, obtained via in-situ annealing of a porous metal-organic framework (MOF), ZIF-8, working at 300°C, gave a resistive signal for ethanol and acetone with a ratio of 1.87 [4]. Gardner et al deposited an Au-SnO2 nanocomposite on a CMOS-MEMS platform for the detection of ethanol at 400°C, the resistive signal for ethanol was 3.5 times higher than that of acetone [5]. The main drawbacks of these nanomaterials are their relatively high working temperature and their rather poor specificity. Under visible light illumination, Di Natale et al studied the porphyrin-functionalized ZnO nanorod photoconductivity changes, modulated by exposure to ethanol and trimethylamine. The signal for trimethylamine was found to be 150 times higher than that of ethanol [6]. Other sensors for the detection of ethanol are based on carbonaceous nanomaterials. Using a carbon black/polyvinylpyrrolidone composite chemoresistor, working between 25°C and 55 °C, Gardner et al detected ethanol with a quantification limit of 2270 ppm and an ethanol on methanol  signal ratio of 0.75 [7]. Wilson et al presented a resistor sensor based on reduced graphene oxide (rGO), working at room temperature. The same sensitivity was obtained for ethanol and methanol, this sensitivity being influenced by humidity [8]. Many sensors based on carbon nanotubes (CNTs) were able to detect ethanol. Recently, Shaalan et al [9] obtained, with CNTs grown on 24 nm of Ni, a quantification limit of 5 ppm for ethanol and an ethanol on acetone signal ratio of 5.2.”

  1. Figure with the scheme of sensors’ construction stages should be included.

A flow-chart of the fabrication of the micro conductometric ethanol sensor is presented as Figure 3.

  1. Some Figures should be merged, e.g. Fig 2 and 11 and some Figs with the results.

Figure 2 and Figure 11 were merged as Figure 2A and 2B. Figures 11 and 12 were merged ad Figure 11A and 11B.

  1. What flow conditions (sccm) were used during the sensing experiments? What were the dimensions (size, etc.) of the gas sensing chamber? These parameters can have a significant impact on the sensor signal and were not discussed at all. Samples were heated somehow or measurements were made in STP?

      As explained in §2.2., the sensor is placed in the headspace (volume of the gas exposure chamber: 15 cm3) over the liquid phase, there is no flow of the gas phase. The measurements were made at room temperature (23±1°C) as reported in §2.2.

  1. Sensors drift were evaluated and somehow?

      As shown in Figure 8, there is no sensor drift, the signal is due to the enzymatic reaction in the presence of the specific VOC.

  1. The conclusion section of the article presents the superiority of the proposed method. But highlight the important outcomes obtained from this research which establish that the methodology is better than the previous methods. However, novelty statement of presented studies should be clearly presented also perspectives should be outlined.

      These comments were added:

“The comparison of the selectivity of detection of the present ethanol sensor to that of the ethanol sensors based on ZnO, carbonaceous nanomaterials shows the interest of the fabricated ADH-based sensor with an ethanol signal 8.3 times higher than that of methanol and 160 times higher than that of acetone.”

As perspective: “The prepared ethanol sensor is then of interest for personal use due to its easy use.”

Comments on the Quality of English Language

Expert language editing is obligatory, it should greatly improve the manuscript. It should be proofread and edited to improve the English throughout understand.

The manuscript was totally revised by a native English person.

Reviewer 2 Report

In this paper (nanomaterials-2513820), the authors exhibited an ethanol sensor based on ADH-NAD+-GNP chitosan composite. The topic is interesting for the target readers and the results are basically acceptable. However, there are some problems in the motivations, experimental details, data results and discussion. As such, major revisions are needed before possible publication.

1.         Title: The title is too long; it is recommended to shorten it.

2.         Abstract: Suggest providing key data results.

3.         Introduction: (1) Many statements without reference support are unscientific, for example, in the first and second paragraphs. (2) Some statements are inconsistent with the cited literature. For example, references 4 and 5 are not typical studies on ethanol gas sensors. In fact, there are many materials related to ethanol gas sensors, such as ZnO, SnO2, and LaFeO3 nanotubes. (3) The current research status of ethanol sensors lacks discussion, and providing only the types of materials is not enough. It is necessary to combine performance data and method strategies to systematically analyze the research progress of ethanol sensors, identify existing problems, and highlight the innovation of this study.

4.         Materials and methods: (1) The material source needs to be explained. (2) How are different gas concentrations achieved? Do you consider the influence of humidity? (3) The testing and parameter definition of sensors should be placed after material and sensor preparation. (4) The definition of gas sensor (response recovery time, response, sensitivity, etc.) needs to be provided and supported by references, may refer to Mater. Chem. Phy. 302 (2023) 127768.

5.         It is recommended to unify the unit expression, and gas concentration is usually expressed in ppm, ppb.

6.         Figure 8 and Table 2: The response has not reached saturation, which will affect the evaluation of response and recovery times.

7.         Do the ethanol sensor proposed in this work has performance advantages? Suggest comparing with recent reports.

8.         The systemic gas sensing mechanism is missing in this work.

9.         The figures’ quality needs to be improved, such as line width, font size, and resolution, especially in Figures 3, 6-10, 12. Additionally, the figures need to be combined reasonably.

10.     References list: most of references are out of date (before 2012). The hot research field of gas sensors is developing rapidly, and it is recommended to cite the latest literature.

11.     Check English writing and journal format. For example, the numbers in the chemical formula require subscripts, including references.

Minor editing of English language required

Author Response

Manuscript ID: nanomaterials-2513820

Title: A sensitive micro conductometric ethanol sensor based on an alcohol dehydrogenase-gold nanoparticle chitosan composite.

Responses to reviewer 2’ comments

The authors thank the reviewer for valuable comments that will improve the quality of the manuscript.

Reviewer 2

In this paper (nanomaterials-2513820), the authors exhibited an ethanol sensor based on ADH-NAD+-GNP chitosan composite. The topic is interesting for the target readers and the results are basically acceptable. However, there are some problems in the motivations, experimental details, data results and discussion. As such, major revisions are needed before possible publication.

  1. Title: The title is too long; it is recommended to shorten it.

The title was shortened as follows: “A sensitive micro conductometric ethanol sensor based on an alcohol dehydrogenase-gold nanoparticle chitosan composite”

  1. Abstract: Suggest providing key data results.

The abstract was completed with key data results: “A microconductometric sensor was designed, based on a chitosan composite including alcohol dehydrogenase and its cofactor and gold nanoparticles, and was calibrated by differential measurements in the headspace of aqueous solutions of ethanol. The role of gold nanoparticles (GNPs) was crucial for improving the analytical performance of the ethanol sensor in terms of response time, sensitivity, selectivity, and reproducibility. The response time is reduced to 10 s, compared to 21 s without GNPs. The sensitivity is 416 µS/cm (v/v%)-1 which is 11.3 times higher than without GNPs. The selectivity factor versus methanol is 8.3, three times higher than without GNPs. The relative standard deviation (RSD) obtained with the same sensor is 2%, whereas it was found to be 12% without GNPs. The measurement of ethanol in the operator’s mouth shows a very high content of ethanol, just after rinsing with an antiseptic mouthwash containing 42% of ethanol and a background level was reached only after water rinsing.

  1. Introduction: (1) Many statements without reference support are unscientific, for example, in the first and second paragraphs. (2) Some statements are inconsistent with the cited literature. For example, references 4 and 5 are not typical studies on ethanol gas sensors. In fact, there are many materials related to ethanol gas sensors, such as ZnO, SnO2, and LaFeO3 (3) The current research status of ethanol sensors lacks discussion, and providing only the types of materials is not enough. It is necessary to combine performance data and method strategies to systematically analyze the research progress of ethanol sensors, identify existing problems, and highlight the innovation of this study.
  • The first part of the introduction was shortened as follows:

“The mouth, due to its role as an interface between the outside world and the interior of the body receiving food, is exposed to multiple potentially pathogenic agents. The active ingredients in antiseptic mouthwashes must be effective against bacteria and fungi, particularly of the Candida type. Their antifungal action is indeed required to prevent oral candidiasis. These antiseptics are either chlorhexidine, cetylpyridinium chloride, hexetidine, hydrogen peroxide, sodium benzoate, or povidone iodine. The main role of ethanol is to solubilize substances responsible for flavor, or active molecules, to increase their bioavailability. Following the positive alcohol test - in the workplace, by the employer - of an employee who denied any consumption, the occupational health service was asked about the possible interference of a mouthwash containing alcohol taken in the minutes preceding breath alcohol measurements. The permitted level is 0.25 mg/L of ethanol in the exhaled air (0.012 v/v% or 120 ppm) which should correspond to 0.5 g/L of ethanol in blood in case of ingestion of alcohol [1]. if the test is positive, this can have economic and socio-professional consequences for employees. The data available in the literature support the hypothesis that a product containing alcohol used as a mouthwash may interfere with alcohol measurements in the expired air in the first 15 min [1].”

  • The discussion about ethanol sensors based on nanomaterials was improved as follows:

“Numerous sensors for the detection of ethanol are based on semiconducting metal oxides, as presented in a recent review by Weimar et al [2]. Gardner et al showed that the sensor based on the electrical resistance of ZnO nanorods, working at 350°C, is influenced by humidity; the ethanol on acetone signal ratio was found to be 3.3 [3]. A ZnO nanoparticle network, obtained via in-situ annealing of a porous metal-organic framework (MOF), ZIF-8, working at 300°C, gave a resistive signal for ethanol and acetone with a ratio of 1.87 [4]. Gardner et al deposited an Au-SnO2 nanocomposite on a CMOS-MEMS platform for the detection of ethanol at 400°C, the resistive signal for ethanol was 3.5 times higher than that of acetone [5]. Under visible light illumination, Di Natale et al studied the porphyrin-functionalized ZnO nanorod photoconductivity changes, modulated by exposure to ethanol and trimethylamine. The signal for trimethylamine was found to be 150 times higher than that of ethanol [6]. Other sensors for the detection of ethanol are based on carbonaceous nanomaterials. Using a carbon black/polyvinylpyrrolidone composite chemoresistor, working between 25°C and 55 °C, Gardner et al detected ethanol with a quantification limit of 2270 ppm and an ethanol on methanol  signal ratio of 0.75 [7]. Wilson et al presented a resistor sensor based on reduced graphene oxide (rGO), working at room temperature. The same sensitivity was obtained for ethanol and methanol, this sensitivity being influenced by humidity [8]. Many sensors based on carbon nanotubes (CNTs) were able to detect ethanol. Recently, Shaalan et al [9] obtained, with CNTs grown on 24 nm of Ni, a quantification limit of 5 ppm for ethanol and an ethanol on acetone signal ratio of 5.2.”

  1. Materials and methods:

(1) The material source needs to be explained.

The source of the reagents is given if §2.1; the source of the chip is given in §2.2; the source of the conductometer is given in § 2.2..

(2) How are different gas concentrations achieved? Do you consider the influence of humidity?

The concentration of the gas in the headspace above the liquid phase is calculated from the Henry’s law as explained in §2.5.

(3) The testing and parameter definition of sensors should be placed after material and sensor preparation.

This point was done

(4) The definition of gas sensor (response recovery time, response, sensitivity, etc.) needs to be provided and supported by references, may refer to Mater. Chem. Phy. 302 (2023) 127768.

The reference was added as Ref 25.         

  1. It is recommended to unify the unit expression, and gas concentration is usually expressed in ppm, ppb.

The unit of ppm were used when the concentration were low enough.

  1. Figure 8 and Table 2: The response has not reached saturation, which will affect the evaluation of response and recovery times.

The measurement was stopped after the inflection point, the response time is lightly underestimated whereas the recovery time to the background is well estimated.

  1. Do the ethanol sensor proposed in this work has performance advantages? Suggest comparing with recent reports.

This point was discussed in the conclusion:

“This work shows how the addition of gold nanoparticles can greatly improve the analytical performance of an ADH-based micro conductometric sensor: decrease of the response time by a factor of 2.1, increase of the sensitivity by a factor of 11.3, decrease of the detection limit up to 106 ppm,  relative standard deviation decreased by a factor of 6, a higher selectivity factor (multiplied by 3.2) to methanol. The comparison of the selectivity of detection of the present ethanol sensor to that of the ethanol sensors based on ZnO, carbonaceous nanomaterials shows the interest of the fabricated ADH-based sensor with an ethanol signal 8.3 times higher than that of methanol and 160 times higher than that of acetone.”

  1. The systemic gas sensing mechanism is missing in this work.

The gas sensing mechanism is presented as the enzymatic reaction, in the introduction.

“The enzymatic reaction, leading to the oxidation of ethanol into acetaldehyde and the release of one proton per oxidized ethanol molecule, is as follows:       

CH3CH2OH + NAD+                      CH3CHO + NADH + H+

  1. The figures’ quality needs to be improved, such as line width, font size, and resolution, especially in Figures 3, 6-10, 12. Additionally, the figures need to be combined reasonably.

The quality of figures was improved. Figure 2 and Figure 11 were merged as Figure 2A and 2B. Figures 11 and 12 were merged ad Figure 11A and 11B.

  1. References list: most of references are out of date (before 2012). The hot research field of gas sensors is developing rapidly, and it is recommended to cite the latest literature.

       Some new references were added.

  1. Check English writing and journal format. For example, the numbers in the chemical formula require subscripts, including references.

The manuscript was totally revised by a native English person.

Reviewer 3 Report

This is a nice paper showing how simple addition of GNPs affected response and response time of ADH/conductometric sensors for detecting alcohol in the workplace.

1) I appreciated Table 2 showing response times and LOD for different EtOH sensors. Is there some sense of difference in cost? 

2) It is important to see how durable and reproducible the sensors are - how do they perform over time, how much do results vary between sensors, what happens if they are stored for months on the shelf. etc. Such studies are an important aspect of sensor development.

3) The lack of error bars suggests experiments were run one time with one sensor. Surely that is not the case?

4) The paper describes a human user breathing into the device, but there is no mention of an institutional review board. Is this the experience of a single user? Is the research exempt from review (I don't think so?).

The English was poor. Nearly every sentence had grammatical flaws making it difficult to read. Repetition, for example, in the first sentence of the abstract. Odd translations (bath of mouth for mouthwash?). Incorrect verbs, incorrect commas. It makes the paper unappealing to read.

These significantly detract from what looks like a reasonably interesting paper showing a novel method for an ethanol sensor.

Author Response

Manuscript ID: nanomaterials-2513820

Title: A sensitive micro conductometric ethanol sensor based on an alcohol dehydrogenase-gold nanoparticle chitosan composite.

Responses to reviewer 3’ comments

The authors thank the reviewer for valuable comments that will improve the quality of the manuscript.

Reviewer 3

This is a nice paper showing how simple addition of GNPs affected response and response time of ADH/conductometric sensors for detecting alcohol in the workplace.

1) I appreciated Table 2 showing response times and LOD for different EtOH sensors. Is there some sense of difference in cost? 

It is difficult to estimate the final cost of the published sensors.

As perspective we declare in the conclusion

The prepared ethanol sensor is then of interest for personal use due to its easy use.”

2) It is important to see how durable and reproducible the sensors are - how do they perform over time, how much do results vary between sensors, what happens if they are stored for months on the shelf. etc. Such studies are an important aspect of sensor development.

These points are indicated in §3.3.

The relative standard deviation (RSD) obtained with the same sensor is 2%. The sensor retains its detection sensitivity  for one month, when kept in a fridge at 4 °C between measurements. The inter-sensor reproducibility obtained for 5 sensors was 8%.

3) The lack of error bars suggests experiments were run one time with one sensor. Surely that is not the case?

The reported points are the mean values obtained for 5 measurements with the same sensor. The size of points is equal to the RSD.

4) The paper describes a human user breathing into the device, but there is no mention of an institutional review board. Is this the experience of a single user? Is the research exempt from review (I don't think so?).

The test was performed on different operators and the results were in agreement with what was observed in Ref 1, using a classical alcohol-meter used by the police.

Comments on the Quality of English Language

The English was poor. Nearly every sentence had grammatical flaws making it difficult to read. Repetition, for example, in the first sentence of the abstract. Odd translations (bath of mouth for mouthwash?). Incorrect verbs, incorrect commas. It makes the paper unappealing to read.

These significantly detract from what looks like a reasonably interesting paper showing a novel method for an ethanol sensor.

The manuscript was totally revised by a native English person.

Reviewer 4 Report

Reviewer report: 

The title of this work is fully missleading. Someone would expect an in-depth investigation on the influence of mouthwash solutions on the alcohol exhaled in breath. Unfortunately the application part is extremely weak.

The authors carried out a single experiment using a single mouthwash brand at conditions that are far away from being realistic. What is the meaning of taking the measurement exactly after using the mouthwash?

The authors state that the content of alcohol was 291 higher than thew allowed limit, but this is unrealistic. What is the detected level after 1-2 minutes of normal breathing? I assume that if an individual rinses his mouth with whiskey the levels at "zero time" will be extremely high.

The authors should have carried out their application under real life conditions and should have compared their values with a calibrated alcohol-meter similar to that used by the police. 

Nevetheless the content of ethanol in mouthwashes seems to me unlikely to cause any values higher than the allowed ones (under realistic measurement conditions).  

No comments

Author Response

Manuscript ID: nanomaterials-2513820

Title: A sensitive micro conductometric ethanol sensor based on an alcohol dehydrogenase-gold nanoparticle chitosan composite.

Responses to reviewer 4’ comments

The authors thank the reviewer for valuable comments that will improve the quality of the manuscript.

Reviewer 4

The title of this work is fully missleading. Someone would expect an in-depth investigation on the influence of mouthwash solutions on the alcohol exhaled in breath. Unfortunately the application part is extremely weak.

The authors carried out a single experiment using a single mouthwash brand at conditions that are far away from being realistic. What is the meaning of taking the measurement exactly after using the mouthwash?

The authors state that the content of alcohol was 291 higher than thew allowed limit, but this is unrealistic. What is the detected level after 1-2 minutes of normal breathing? I assume that if an individual rinses his mouth with whiskey the levels at "zero time" will be extremely high.

The authors should have carried out their application under real life conditions and should have compared their values with a calibrated alcohol-meter similar to that used by the police. 

Nevetheless the content of ethanol in mouthwashes seems to me unlikely to cause any values higher than the allowed ones (under realistic measurement conditions).  

The title of the manuscript was changed as follows:

A sensitive micro conductometric ethanol sensor based on an alcohol dehydrogenase-gold nanoparticle chitosan composite.

We agree with the reviewer, after rinsing one’s mouth with whiskey, the results would be the same. Rinsing mouth with a mouthwash is an antiseptic  treatment which is not the case for whiskey. The test with mouthwash was performed on different operators and the results were in agreement with what was observed in Ref 1, using a classical alcohol-meter used by the police.

Round 2

Reviewer 2 Report

The revised manuscript can be published.

Author Response

No revision is required.

Reviewer 3 Report

Thank you, I have no further comments on the science.

Thank you for the new draft, which is much improved. Please review carefully for typos and symbols which should be superscript or subscript (NAD+, SnO2 eg). You define IDEs, so you should use that term and not interdigitated electrodes the rest of the manuscript. Please proofread carefully. Also a few phrases which are too colloquial: "Kept in the fridge for one night" should be replaced with "refrigerated overnight". A drop of sample was deposited, not "the sample was dropped ..." 

Author Response

All the required corrections were done.

Reviewer 4 Report

The revised version could be accepted for publication 

N/A

Author Response

No required correction.